# Lessons on Differential Neuronal-Death-Vulnerability from Familial Cases of Parkinson’s and Alzheimer’s Diseases

**DOI:** 10.3390/ijms20133297

**Published:** 2019-07-04

**Authors:** Rafael Franco, Gemma Navarro, Eva Martínez-Pinilla

**Affiliations:** 1Chemistry School, University of Barcelona, 08028 Barcelona, Spain; 2Centro de Investigación Biomédica en Red Enfermedades Neurodegenerativas (CiberNed), Instituto de Salud Carlos III, 28031 Madrid, Spain; 3Department of Biochemistry and Physiology, Faculty of Pharmacy, University of Barcelona, 02028 Barcelona, Spain; 4Departamento de Morfología y Biología Celular, Facultad de Medicina, Universidad de Oviedo, 33006 Oviedo, Spain; 5Instituto de Neurociencias del Principado de Asturias (INEUROPA), 33003 Oviedo, Asturias, Spain; 6Instituto de Salud del Principado de Asturias (ISPA), 33011 Oviedo, Asturias, Spain

**Keywords:** glucocerebrosidase, maternal imprinting, mitochondria, steryl glycosides, cholesterol

## Abstract

The main risk of Alzheimer’s disease (AD) and Parkinson’s disease (PD), the two most common neurodegenerative pathologies, is aging. In contrast to sporadic cases, whose symptoms appear at >60 years of age, familial PD or familial AD affects younger individuals. Finding early biological markers of these diseases as well as efficacious treatments for both symptom relief and delaying disease progression are of paramount relevance. Familial early-onset PD/AD are due to genetic factors, sometimes a single mutation in a given gene. Both diseases have neuronal loss and abnormal accumulations of specific proteins in common, but in different brain regions. Despite shared features, the mechanisms underlying the pathophysiological processes are not known. This review aims at finding, among the genetic-associated cases of PD and AD, common trends that could be of interest to discover reliable biomarkers and efficacious therapies, especially those aimed at affording neuroprotection, i.e., the prevention of neuronal death.

## 1. Introduction

Parkinson’s disease (PD) and Alzheimer’s disease (AD) are neurodegenerative and have in common that familial cases, i.e., those caused by inheritable genetic factors, display early symptoms and are much more scarce than sporadic ones. Furthermore, the main risk factor in sporadic cases is age. Although the brain areas suffering from neuronal death are different, both diseases show abnormal protein deposits that are visible using a light microscope. In PD, alpha-synuclein deposits, also known as Lewy bodies, appear in the substantia nigra and surrounding brain regions and, for this reason, PD is considered an alpha-synucleinopathy. Other synucleinopathies, such as dementia with Lewy bodies (DLB), have been reported. The brain of AD patients presents two kinds of protein aggregations that are considered the pathological hallmarks of the disease, namely amyloid plaques and neurofibrillary tangles. Plaques result from abnormal processing of the amyloid precursor protein (APP), leading to the aggregation-prone peptide, β-amyloid. Tangles result from the aggregation of aberrantly phosphorylated tau proteins. Interestingly, plaques are extracellular and tangles are found inside neurons. Importantly, the location of alpha-synuclein deposits in the midbrain of PD patients, and of plaques and tangles in the hippocampus and cortex of AD brains, seems intimately linked to the main symptoms of these pathologies (motor alterations and memory loss, respectively). It is known that tau is part of the cytoskeletal machinery involved in axon development and function in neurons. However, it should be noted that the physiological function of alpha-synuclein and of APP has not been yet elucidated, and that plaque presence does not always correlate with cognitive AD-related symptoms [1,2,3]. In fact, treatments that attempt to decrease the amyloid burden, mainly using antibodies for plaque clearance, have not been proven effective in patients due to different reasons [4]. It still remains doubtful whether amyloid deposits (Figure 1) are beneficial (neuroprotective) or if, as per the amyloid hypothesis, the inductors of tau pathology and the main reason for neuronal death and neurodegeneration. The purpose of the present article is to find common trends among the genetic-associated cases of PD and AD, i.e., to provide hints to know i) why some neurons are more vulnerable than others, and ii) whether there are common mechanisms of neuronal death in patients inheriting mutations that lead to early-onset PD or early-onset AD. Examples have been selected to provide a general overview, but an exhaustive list of mutations linked to PD or to AD may be found in very recent reports [5,6,7,8,9,10]. Here, we consider familial PD and familial AD, i.e., we do not consider cases such as familial AD developing sporadic PD (or vice versa).

## 2. Mutations Leading to PD but Not to AD

The gene *GBA1* encoding for β-glucocerebrosidase 1 (GBA1) was already associated with PD in 1996 [11,12]. However, interest in such a discovery has recently re-emerged. The renewed attention has likely been triggered by the possibility of developing virus-based gene therapies in the near future. The majority of mutations in the gene lead to reduced GBA1 enzyme activity and the well-known Gaucher’s disease (GD). This highly prevalent lysosomal storage disorder (LSD) leads to an abnormal accumulation of glycolipids in different body tissues causing a variety of symptoms, probably related with the remaining activity displayed by the mutant enzyme (anaemia, bone pain, fractures, and, eventually, neurological alterations). Although GD primarily affects peripheral tissues, recent observations have shown that heterozygous mutations in the *GBA1* gene represent the most important genetic risks, not only in PD but also in other synucleinopathies (LBD included) [13,14]. As an LSD, like the Tay-Sachs’ or the Niemann-Pick’s diseases, the need to find ad hoc treatment for patients was characterized long ago. Neurological symptoms were overlooked or considered of less relevance than others resulting from abnormal lysosomal function. Indeed, patients show an accumulation of glucocerebrosides, whose degradation route starts by removing the glucose moiety to provide ceramides, which can be converted into sphingoid bases (the best known is sphingosine) for reuse (see [15] for review).

There is consensus that reducing GBA1 activity leads to alpha-synuclein aggregation, but the mechanism is not known. On the one hand, the possibility that GBA1 is able to disrupt aggregates does not fit with the nature of GBA1 catalysis, which is not proteolytic. Accordingly, it is not expected that GBA1 would directly act on amyloid plaques or neurofibrillary tangles. To our knowledge, no reliable data has linked mutations in the gene to early-onset inheritable AD. On the other hand, solid biochemical evidence indicates that the metabolism of cholesterol, and in particular of steryl glycosides, may be involved in neuroprotection/neurodegeneration events. Many cells have a second β-glucocerebrosidase, GBA2, that shares transglycosylation activity with GBA1 [16,17]. Hydrolysis versus transglycosylation balance keeps steryl glycosides at appropriate levels. In healthy conditions, GBA1 promotes the degradation of β-glucosyl-cholesterol (GlcChol) while GBA2 works in the opposite direction, GlcChol synthesis. Due to the fact that imbalance in GlcChol production/degradation leads to neurotoxicity, we have proposed that steryl glycoside levels should also be considered to assess their impact on neurodegenerative processes and, also, to screen for potential mutations in the gene encoding for GBA2 (Figure 2) (See [17] for review). It is, however, puzzling why the imbalance due to *GBA* mutations is more detrimental in just a specific region of the brain (substantia nigra, in the case of PD).

Interestingly, the Pink1/Parkin pathway, intended to maintain mitochondrial integrity, is modulated by a kinase whose activity is regulated by GlcChol, protein kinase B (also known as Akt). This result complements findings of mitochondrial alterations that are relevant for neuronal death in PD.

To assess the differential death vulnerability of neurons depending on the disease, one would need to know (i) the physiological role of the proteins directly involved in the pathology, and (ii) the differential expression of proteins in the different brain regions and neuronal types, i.e., the human brain proteome. In the case of PD, the physiological role of GBA1 is well known, due to mutations in the gene encoding for it. The above comments on the second enzyme’s activity may guide the discovery of biomarkers and the development of de novo therapeutic avenues. Recent data bridges the gap concerning where the enzyme is expressed along the central nervous system (CNS) of a non-human primate, *Macaca fascicularis*. Might this mapping provide clues as to why nigral neurons are more vulnerable than other neurons? The first finding in the seminal work by Lanciego and co-workers (2018) [18] was that GBA1 is not ubiquitously expressed, as one might deduce for being a “lysosomal enzyme” and for being necessary to degrade glucocerebrosides, which are present in virtually all cells. A second important finding was a cell-to-cell differential expression. The authors found very high levels of the enzyme in some specific neuronal phenotypes. Finally, a third relevant finding was the uneven expression of GBA1. As the authors indicate, enriched expression of GBA1 was found “*in cholinergic neurons from the nucleus basalis of Meynert, dopaminergic cells in the substantia nigra pars compacta, serotoninergic neurons from the raphe nuclei, as well as in noradrenergic neurons located in the locus coeruleus*” [18]. These results provide an anatomical correlate of differential vulnerability due to specific (and high) expression of the enzyme in the dopaminergic neurons of the substantia nigra. Therefore, GBA1 must have a relevant function, but why are dopaminergic neurons less resistant to reduced GBA1 activity than neurons of other areas in which the enzyme is also heavily expressed?

From the available data, the increased vulnerability of dopaminergic neurons expressing mutated GBA1 may be due to the expression of another protein whose physiological role is regulated by glucocerebrosidase activity. The altered functionality of such a protein would, directly or indirectly, affect mitochondrial function that, in the long run, would lead to cell death. Might the alpha-synuclein protein be affected by GBA1? There are a significant number of papers that suggest such a link. Although the underlying mechanisms are not appropriately deciphered, one hypothesis considers, as previously mentioned, that GBA1 might act directly on Lewy bodies to clear them [19]. This mechanism is not based on evidence, as it would require proteolytic action of the main protein in the aggregates, but no proteolytic activity has been reported for an enzyme with a dual role (glycosidase and glycosyl transferase). Other mechanisms that do not follow Occam’s razor and that may occur in any neuron (not only in dopaminergic neurons) have been suggested [20]. A recent report showing that glucosylceramide (GlcCer) may affect aggregation of alpha-synuclein could be valuable [21], but only if alpha-synuclein is enriched in dopaminergic neurons. Otherwise, this interesting observation would not address the question of how reduced GBA1 activity leads to death of dopaminergic neurons but not of other neurons also expressing GBA1 and alpha-synuclein. For a detailed study of the expression of alpha-synuclein in different brain regions of a mouse, see [22] and references therein.

## 3. Clues Revealed by Reports on Leucine-Rich Kinase 2 (LRRK2) and Parkin

The leucine-rich kinase 2 (*LRRK2*) gene encodes for an enzyme previously known as dardarin, which phosphorylates a variety of proteins at Ser/Thr residue. Hernández et al. (2005) showed that the G^2019^S LRRK2 mutant was associated to PD but not to other neurodegenerative conditions, including essential tremor, late-onset AD, and frontal temporal dementia, a tauopathy affecting the brain cortex [23,24]. Unfortunately, the transgenic animal lacking expression of LRRK*2* has not provided much information on the mechanism(s) leading to increases in alpha-synuclein expression and Lewy bodies formation. The most attractive feature of the model is nigral neurodegeneration, although it occurs at a low pace and the number of affected dopaminergic neurons (22% at 15 months of age) is limited [25]. Like other cell Ser/Thr kinases, LRRK2 has several (potential) protein substrates. A proteomics study performed in neuroblastoma cells, either expressing LRRK2 and pharmacologically treated with an enzyme inhibitor or expressing the G^2019^S LRRK2 mutant, led to identifying 776 phosphorylation sites whose levels changed by +/- 50% upon inhibition of the enzyme. Ingenuity pathway analysis (IPA) of the results suggested that the function of the main substrates of LRRK2 is related with cell morphology and proliferation, and is mediated by regulation of oestrogen, integrins, and extracellular signal-regulated kinase 5 (ERK5) signalling [26]. Alterations due to the expression of the G^2019^S LRRK2 mutant correlate with increased energy demand. Cells in a *Drosophila melanogaster* model showed, among others, apoptosis and mitochondrial disorganization [27]. Ng et al. (2009) [28] demonstrated that transgenic expression of Parkin (*PARK2*) in the fly-based model protected against neurodegeneration triggered by G^2019^S LRRK*2*. A few years earlier, Smith et al. (2005) [29] showed that LRRK2 interacts with Parkin. Remarkably, some mutations in the *PARK2* gene are associated with familial PD. The first evidence for this association was presented in the study of Kitada et al. (1998) [30], which discovered that a gene in chromosome 6 (6q25.2-q27) was associated with autosomal recessive early-onset PD; the study named the gene product “Parkin” [31,32]. There is consensus on the enzymatic action of Parkin as an ubiquitin ligase and its involvement in protein processing degradation. It is, however, not clear whether the altered enzymatic activity in mutant proteins contributes to the disease, i.e., the protein may have other functionalities. In this sense, multiple evidence links Parkin to mitochondrial function. For instance, a recent lipidomic analysis shows age-dependent alterations in the lipid composition of mitochondrial membranes in Parkin knockout mice [33]. Another earlier study showed a correlation between mitochondrial abnormalities and the degeneration of dopaminergic neurons due to the expression of a R^275^W Parkin mutant in a fly model [34]. Further evidence of the involvement of Parkin in mitochondrial function may be found in the review by Truban et al. (2017) [35].

## 4. Main Mutations Leading to Inheritable Early-Onset AD but Not to PD

Here we will focus on the mutations found in AD patients that are displayed by the so-called triple transgenic mouse model of AD (3xTg-AD), an animal showing early cognitive deficits. Although those mutations have been the most studied in the field of AD research, they can only explain a small percentage of familial cases of AD. Notably, many of the genes that potentially produce early-onset AD have not been identified [36]. Some candidates have been suggested, but, to our knowledge, many of them are not fully validated [37]. Preclinical AD research relies on transgenic mice overexpressing human proteins carrying mutations identified in early-onset AD cases. The mouse, also known as Laferla mouse [38], overexpresses the following mutant proteins found in familial AD: the Swedish mutations in the APP (APP_Swe_) [39,40], the P^301^L mutation in tau [41], and the M^146^V mutation in presenilin 1 [42,43,44,45,46]. A summary of the main trends associated with these three AD-linked mutations is found in (https://www.alzforum.org/).

There is controversy on the possible involvement of LRRK2 on AD pathophysiology. On the one hand, the first hint about this potential link was suggested more than a decade ago, but the analysis of 754 patients who meet the Stroke–AD and Related Disorders Association and National Institute of Neurological and Communicative Disorders criteria for AD, did not have the G^2019^S LRRK2 mutation [47]. On the other hand, a direct interaction has been demonstrated between LRRK2 and the APP, since its intracellular domain (AICD) may be phosphorylated by the kinase. In fact, this phosphorylation regulates the transcriptional activity of the AICD in the nucleus after APP is processed by specific proteases [48,49].

The K^670^M/N^671^L APP_Swe_ does not seem to have anything in common with PD, in the sense that abnormal processing of APP_Swe_ leads to β-amyloid production and plaque accumulation but neither to neurodegeneration in the substantia nigra nor to PD-like motor symptoms. In summary, the pathological mechanism in patients with this mutation is the aberrant processing of the type I membrane protein by ad hoc proteins, secretases, leading to a β-amyloid peptide which, being water insoluble, tends to aggregate (see [50] for review). It should be noted that a proteomics study using the Tg2576 mouse model, which overexpresses APP_Swe_, shows marked alterations in mitochondrial protein expression [51] (see below). In fact, there is preclinical evidence of a mitochondria-neurodegeneration link as a common trend in familial AD and PD cases.

It is shown that the Tg2576 transgenic mice overexpress a voltage-dependent anion-selective channel (VDAC1), which is located in the outer mitochondrial membrane and regulates ATP transport/metabolism. One alteration related with this channel was hyperphosphorylation at a specific glycogen synthase kinase-3β site, an enzyme whose activity was also increased. In addition, hexokinase I, a key enzyme in energy metabolism that is able to interact with VDAC1, was altered [52]. These findings are relevant since post-mortem samples from AD patients at advanced stages of the disease showed an overexpression of VDAC1. To further identify proteins whose aberrant expression in the transgenic model could have a role in neuronal degeneration, a proteomics study was undertaken. First of all, it was noticed that the percentage of affected proteins was highest in the case of mitochondria. As an example, an enzyme presence in the mitochondrial matrix (which was not previously related to AD), propionyl-CoA carboxylase, was deregulated. Secondly, altered mitochondrial functionality appeared before cognitive symptoms and involved increased cytochrome c oxidase activity [51]. Overall, the differential proteome in this mouse expressing a mutated form of the human APP showed mitochondrial and metabolic alterations that could compromise the fate of the affected neurons. What is interesting is that cognitive impairments, which appear late in this animal, run their course without neuronal death. It is worth mentioning that recent results from our laboratory suggest the occurrence of neuroprotective microglia in transgenic models of AD [53]. Accordingly, it would be of interest to analyse, in post-mortem samples of AD and PD patients, the most predominant microglial phenotype M1 or proinflammatory, or M2 or neuroprotective. The possibility to modify, by pharmacological means, the phenotype of microglia is attracting attention in the neurodegenerative disease research field [54]. Again, a key factor is age, meaning that mitochondria are completely functional for years before they fail to fulfil the energy requirements of neural events.

Also relevant is the finding (in the same Tg2576 model) of a maternal imprinting that confers a greater facility to launch an AD-like neurodegenerative cascade in wild-type animals. Obviously, the alteration in such scenario comes from the transmission of the mitochondria of mothers who overexpress mutant APP_Swe_ [55]. It is evident that the pathological findings in the progeny can only come from already altered mitochondria of mothers who do not exhibit any cognitive deficits.

The majority of familial AD cases are due to mutations in presenilin genes [56,57]. Presenilins are a family of membrane proteins with atypical structures. They contain a N-terminal domain, a C-terminal domain, and span the lipid bilayer several times (between 7 and 9 times [58]) but with little resemblance to the best-known superfamily, the heptaspanning G-protein-coupled receptor superfamily. Their physiological function is not fully established, but there is consensus in considering them a part of the secretase complex involved in APP processing. Accordingly, mutations in presenilin genes would lead to abnormal APP processing, which would explain its link to early-onset AD [36,37]. The marked efforts to understand the function of presenilins have led researchers to suggest their involvement in a myriad of cellular events: calcium homeostasis, Wnt/β-catenin signalling, apoptosis, and protein trafficking and degradation (see [58] for review). This multifunctional role attributed to presenilins prevents finding any specific relationship with events taking place in familial PD patients. However, assuming that mitochondria has an important role in some AD and PD cases, it is worth highlighting that presenilins regulate mitochondrial calcium homeostasis in the hippocampal mossy fiber pathway, a complex synaptic network between the dentate gyrus and CA3 area, crucial in learning and memory processes [59].

Neurodegeneration with tau involvement is intriguing. Although familial cases exist, they are the minority if compared with neurodegenerative diseases coursing with aberrant tau hyperphosphorylation and intracellular deposition in structures known as neurofibrillary tangles. As the deposition of alpha-synuclein aggregates has given its name to synucleinopathies, diseases involving the formation of aberrant tau and tau-containing deposits are known as tauopathies, which include Pick’s disease, supranuclear palsy, and corticobasal degeneration. Neurofibrillary lesions may be linked to mutations in the tau gene but also to mutations in other genes. However, the majority of reported tauopathies are not associated with any specific gene mutation [60].

## 5. Network and Cell Function Analyses

Attempting to obtain more information from the genes that cause familial AD or PD, we used the STRING (Search Tool for the Retrieval of Interacting Genes/Proteins) database (https://string-db.org/) to look for interconnections between the products of the above-commented genes. STRING is a free-to-use on line tool to find correlations useful in system biology approaches. The results obtained just using the main genes associated to familial AD/PD are shown in Figure 3A. The node connecting the AD-related and PD-related genes was APH1B, which is part of the gamma-secretase complex that contains presenilins 1 and 2. GBA2 is only connected to GBA (GBA1), but this is likely due to the few number of papers related to GBA2. This image provides, in our opinion, a significant finding namely that tau (MAPT) is a node connecting PD and AD-related gene products. We repeated the procedure but included products of genes that were described elsewhere as “common and rare variants associated to AD” [37]. The results are shown in Figure 3B. Although the picture is similar, a new link that appears as the node of Apo E is not only connected to AD-related genes/proteins but also to alpha-synuclein and GBA1, and even to GBA2. Another interesting finding is the lack of connection between the four AD-related proteins: CELF6 (CUGBP Elav-Like Family Member 6), CR1L (Complement C3b/C4b Receptor 1 like), FERMT1 (fermitin), ABCA8 (ATP Binding Cassette Transporter), UNC5C (UNC-5 family of netrin receptors), and CLUL1 (clusterin). This may result from the lack of relevance of these genes or from the need for more studies to assess their relevance.

In terms of cell function, STRING provides little information. Thus, there has been no substantial improvement respective to the studies performed in 2007 to address the differential transcriptome of PD and AD using samples from patients. The authors of the study highlighted that the altered transcripts encoded for proteins involved in APP processing, synaptic vesicle handling, and the function of receptors of the insulin family; furthermore, they highlighted that a significant amount of proteins were related to oxidative stress [61]. In that sense, we must remark that mitochondria are at the center stage concerning oxidative stress.

## 6. Remarks and Conclusions

If any, there is little overlapping between genes whose mutations (or deletions) cause PD and AD. This fact reflects that mutated genes have differential functionality among brain regions, probably due to their differential expression. In contrast to other monogenic alterations, which are evident at birth or at young ages, familial PD or AD patients present symptoms much later in life. For instance, whereas GD due to mutations in the *GBA1* gene is usually diagnosed soon after birth, GBA1-associated PD symptoms appear much later in life.

Once again, age is the first key factor to consider. Even the most noxious mutation leading to early-onset AD or PD provides clinical symptoms after decades of life. This contrasts with the myriad of monogenic-bases diseases that lead to symptoms soon after birth. The other two common factors pointed out here are mitochondrial deficits that, in the long run, contribute to neuronal death and metabolic defects in the sphingolipid/sterol interface. Unfortunately, there is not enough data to ascertain whether mitochondrion-related metabolites or sterol-derivatives in plasma or cerebrospinal fluid (CSF), may be biomarkers for better diagnosis or for better management of neuroprotective interventions.

Familial PD is a different disease from late-onset PD. Likewise, familial AD is a different disease from late-onset AD. However, familial cases may provide clues about the causes of neuronal death. One of these causes is the verification that neuronal death does not require abnormal extracellular deposits, such as plaques. In fact, neuronal death occurs by intracellular alterations in neurons due to abnormal metabolism and/or lack of vital support from surrounding glia. This fact is confirmed by the several cases of plaque deposition (found in necropsies) without any symptom of AD-related or AD-unrelated dementia. It seems that increases in the number of senile plaques and reduced levels of 1-42 β-amyloid (Aβ_1-42_) peptide in the CSF correlate with brain atrophy in non-demented elderly [62,63,64]. Interestingly, a plaque burden was associated with depression in aged individuals enrolled in the “Mayo Clinic Patient Registry or Study of Aging”, whose clinical dementia score was zero [64].

Intracellular protein aggregates occur in both familial and sporadic AD and PD cases. It is suggested that gene therapy with viruses/vectors containing the sequence encoding for GBA1 may clear alpha-synuclein aggregates. Apart from the fact that GBA1 has no proteolytic activity, it remains to be elucidated whether clearing Lewy bodies or neurofibrillary tangles will lead to similar failures as those obtained in the case of therapies addressed to clear β-amyloid-containing plaques in AD patients [4,65,66]. Is there any common trend in familial AD/PD that could be of interest in drug discovery approaches? Reports, at the very least, provide two hints: one related to alteration of mitochondrial function and another related to cholesterol metabolism.

It should be noted that mitochondrial alterations are common in PD/AD, both familial and sporadic. They do not only lead to less ATP production but to calcium homeostasis alterations, loss of cell membrane integrity, and mitophagy, among others. For specific mechanisms relating mitophagy to neurodegenerative diseases, see the recent report by Wang et al. (2019) [67]. Apart from the above-described cases, there are more cases that directly demonstrate the PD-linked mutation in the *ATP13A2* gene, which codes for a protein involved in calcium homeostasis, and whose deficit in neurons leads to mitochondrial and lysosomal dysfunction [68].

Cholesterol metabolism is very important in mammals. Concerning PD, the huge knowledge on GBA1 catalytic functions places the focus on the metabolism of steryl glycosides, molecules that are known to trigger a PD-related synucleinopathy in rodents. A little-known model of PD consists of the administration of β-sitosterol β-D-glucoside. This is a progressive model of the disease that has advantages over other more known and more commonly used animal PD models. In fact, a differential trend in this model is the accumulation of alpha-synuclein-containing Lewy bodies [69,70]. Apart from an abnormal lysosomal function affecting alpha-synuclein degradation, it is tempting to speculate that toxicity in dopaminergic neurons may be due to the accumulation of steryl glycosides (Figure 2). Reduced activity of GBA1 increases glucosylated lipid, cholesterol and cholesterol-ester levels [71,72]. Even fibroblasts from PD patients with the N^370^S GBA1 mutation accumulate cholesterol [73]. Last but not least, one domain of LIMP-2 (lysosomal integral membrane protein-2) may directly interact with GBA1, allowing traffic from the endoplasmic reticulum to lysosomes, thus regulating sphingolipid processing [74].

Although deciphering the physiological role of steryl glycosides in humans requires further experimental effort, technical advances have allowed us to identify and quantify these molecules in mammals [75]. It is both feasible and useful to look for mutants of the *GBA2* gene in PD and for alterations in the steryl glycoside levels and metabolism in AD. Remarkably, not only GBA1 but GBA2 participates in keeping an appropriate steryl glycoside production/degradation balance in the healthy brain. Moreover, sterol metabolism provides a strong link to Apolipoprotein E (Apo E) [76,77], which is implicated in cholesterol transport. The *APOE* ε4 allele is, indeed, a risk factor for early-onset and late-onset AD. In physiological conditions, Apo E is synthetized in the CNS by astrocytes and facilitates lipid transport to neurons via cell surface receptors. Solid evidence shows that Apo E contributes to β-amyloid metabolism, aggregation, and deposition to form senile plaques in AD brains [76,77,78]. Specifically, it seems that *APOE* ε4 allele carriers show increased β-amyloid plaque deposition [79]. Moreover, it should be noted that Apo E may regulate inflammatory cascades that negatively impact on the pathology [80].

Still puzzling is why similar alterations in mitochondrial function, protein trafficking, and steryl glycoside metabolism lead to neuron death in the substantia nigra in familial PD but not in familial AD. Despite the fact that neurons are the main focus in AD/PD, especially in familial cases, glia-focused research has been consistently undervalued. On the one hand, some of the mutated proteins are expressed in glia. On the other hand, neuroprotective interventions should contemplate glia, for instance in PD, which is diagnosed when >70% dopaminergic neurons have disappeared. Targeting neuroprotective glia may be a better option than targeting the remaining dopaminergic neurons whose death is expected to occur irremediably in a progressive manner.

## Figures and Tables

**Figure 1 ijms-20-03297-f001:**
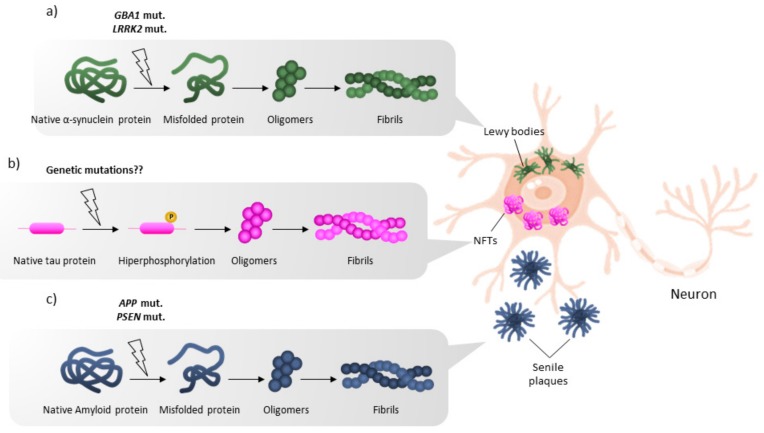
Scheme of pathways involved in the formation of the abnormal protein aggregates that characterize neurodegenerative diseases. (**a**) Misfolded alpha-synuclein proteins form oligomers and fibrils that aggregate into Lewy bodies in Parkinson’s disease (PD) brains and other synucleinopathies. (**b**) Aberrant hyperphosphorylation of tau leads to the formation of oligomers and fibrils that aggregate as pathological intracellular neurofibrillary tangles in Alzheimer’s disease (AD) brains. (**c**) Misfolded amyloid proteins form oligomers and fibrils that aggregate as pathological extracellular senile plaques in AD brains. NFTs: neurofibrillary tangles.

**Figure 2 ijms-20-03297-f002:**
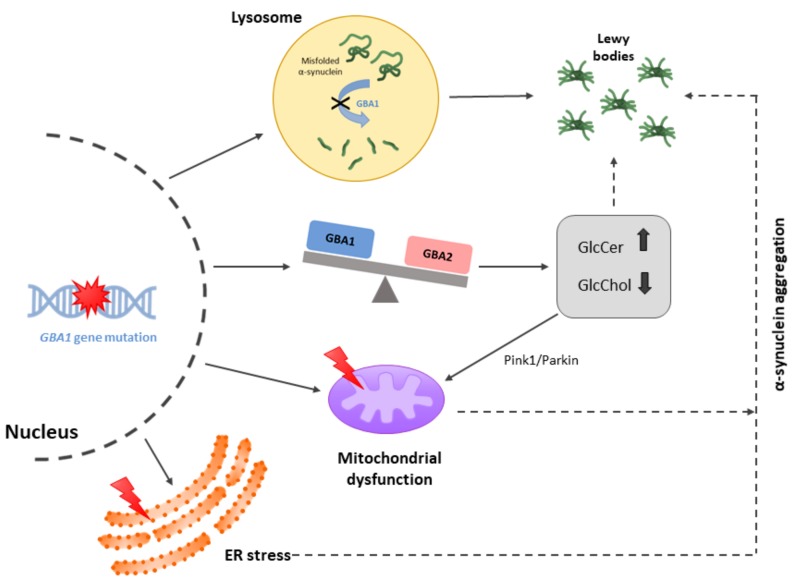
Scheme of the neuropathological processes by which the β-glucocerebrosidase 1 (*GBA1*) gene mutation and reduced GBA1 activity promote alpha-synuclein aggregation with consequent Lewy bodies formation. These mechanisms include impaired lysosomal degradation of misfolded alpha-synuclein proteins, unbalanced GBA1/GBA2 enzyme activity, endoplasmic reticulum (ER) stress and mitochondrial dysfunction. GlcCer: glucosylceramide, GlcChol: β-glucosyl-cholesterol. The red ray depicts altered functionality in the organelle (mitochondria or ER).

**Figure 3 ijms-20-03297-f003:**
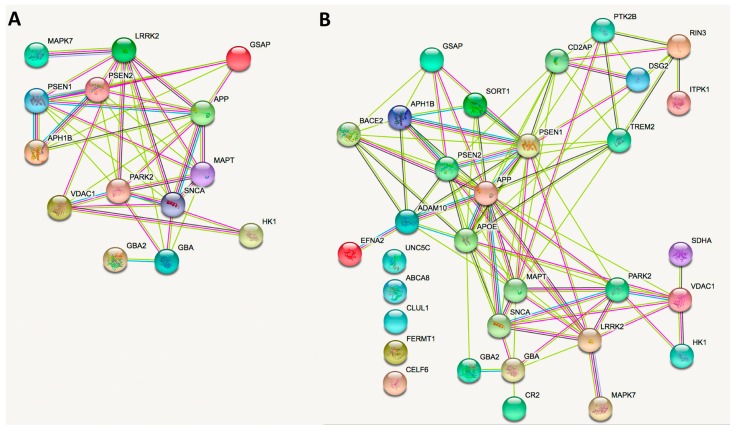
STRING-based network analysis of protein products of genes involved in familial AD or PD. Proteins in panel **A**: alpha-synuclein (SNCA), amyloid precursor protein (APP), GBA1, GBA2, extracellular signal-regulated kinase (MAPK7), hexokinase (HK1), leucine-rich kinase 2 (LRRK2), presenilin (PSEN1 and PSEN2 = APH1B), β-secretase (BACE2), gamma-secretase (GSAP), mitogen-activated protein kinase 7 (MAPK7), succinate dehydrogenase (SDHA), Tau (MAPT) and voltage-dependent anion-selective channel (VDAC1). Parkin1/2 not found in a STRING database for humans. Proteins in panel **B**: those in A plus those described elsewhere [37]: metalloproteinase domain-containing protein 10 (ADAM10), Apolipoprotein E (APOE), ATP Binding Cassette (ABCA8), β-secretase (BACE2), CD2 associated protein (CD2AP), clusterin (CLUL1), complement receptor 2 (CR2), CUGBP (CUG triplet repeat, RNA binding protein 1), Elav-Like (CELF6), desmoglein 2 (DSG2), ephrin receptor (EFNA2), fermitin (FERMT1), inositol polyphosphate-5-phosphatase (ITPK1), protein tyrosine kinase 2 (PTK2B), Ras And Rab Interactor 3 (RIN3), sortilin (SORT1), triggering receptor expressed on myeloid cells 2 protein (TREM2) and Unc-5 netrin Receptor (UNC5C). The colors of the edges represent the different types of protein associations, either from known or predicted interactions: experimentally determined (magenta) and/or from curated databases (blue), protein homology (clear blue), text mining (green), and/or co-expression (black).

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
