# Peer review of "Lessons on Differential Neuronal-Death-Vulnerability from Familial Cases of Parkinson’s and Alzheimer’s Diseases"

_ijms, 2019, doi:10.3390/ijms20133297_

Round 1
Reviewer 1 Report
I think, this is an interesting study. Pathological overlap between neurodegenerative diseases is a hot topics. There may be a common pathway of neurodegeneration, and genetics plays a significant role in neurodegenerative diseases.
I have a few suggestion
LRRK2 was suggested to be an AD risk factor too (even though conflicting results are available on it). . For example PMID: 16401756, PMID 28720718 or PMID 28901623. Could you discuss about this?
Could you discuss a little more about APOE E4 allele and the putative role in neurodegeneration?
A figure on gene interactions and putative common pathways of AD and PD-related genes, and their possible role in neurodegeneration would be interesting.
Author Response
Reviewer 1
I think, this is an interesting study. Pathological overlap between neurodegenerative diseases is a hot topic. There may be a common pathway of neurodegeneration, and genetics plays a significant role in neurodegenerative diseases.
Answer: thanks for the positive comment.
I have a few suggestions:
LRRK2 was suggested to be an AD risk factor too (even though conflicting results are available on it). For example, PMID: 16401756, PMID 28720718 or PMID 28901623. Could you discuss about this?
Answer: we appreciate the suggestion that has been taken into account in the revised version of the manuscript.
Could you discuss a little more about APOE E4 allele and the putative role in neurodegeneration?
Answer: we appreciate the suggestion about APOE E4 allele that has been incorporated, together with 2 new references, to the revised version of the manuscript.
A figure on gene interactions and putative common pathways of AD and PD-related genes, and their possible role in neurodegeneration would be interesting.
Answer: we really appreciate the suggestion. A new figure has been added to the revised version using the STRING tool.
Observation: major changes are highlighted in yellow. Grammatical corrections are not highlighted.
Reviewer 2 Report
The article summarizes the common trends among the genetic-associated cases of PD and AD. The authors indicate that these similarities could be of interest to find reliable biomarkers and efficacious therapies. Throughout the manuscript, the authors give some data, obtained from different research articles, in order to answer two questions: i) why some neurons are more vulnerable than others? and ii) are there common mechanism of neuronal death in patients inheriting mutations that lead to early-onset PD or early-onset AD?
The topic of the article covers the aim of the Special Issue "Genetics of Neurodegenerative Diseases: Focus on Progression and Response to Treatment”, presenting a cohesive picture of the state-of-the-art in the field, and helping to advance in understanding and management of neurodegenerative diseases. The manuscript describes a novel viewpoint and perspective, looking for similarities and differences in genetic between early-onset PD or early-onset AD.
Although it is a valuable article, authors should provide the below changes:
1. Reference numbers should be placed in square brackets [], as it is indicated the Instructions for Authors of the Journal.
2.Size of the words in the draws of Figure 1 and 2 should be increased to improve the quality of the figures.
3. In line 209, abbreviation should be included: “The K670M/N671L Swedish mutations (APPSwe) of the amyloid precursor…”, to define it for the first time because it is used afterwards.
4. In line 220, it should be clarified that in Tg2576 has been observed an overexpression of VDAC1 in hippocampus. Because in the present form, it seems that the transgenic mouse has been generated by the overexpression of VADC1 gene.
5. In line 244, reference should be corrected, the correct is [50].
6. In line 288, the assertion is controversial. Authors should be clarified if this is an own opinion or it is a scientific convention, in which case authors must provide bibliographical references.
7. All the references should be noted as it is indicated in the guide for authors: 1. Author 1, A.B.; Author 2, C.D. Title of the article. Abbreviated Journal Name Year, Volume, page range. Available online: URL (accessed on Day Month Year).
8. In line 469, ref. 32, it should be indicated [Epub ahead of print].
9. In line 514, ref. 46, there is a typo, repetition of J.F.J.F.
10. In line520, ref. 47, there is a typo: “A., M., C.-T., M., V., F., C., J., D. R., and D., F.”
11. In line 539, ref. 54, reference should be corrected: Oncotarget. 2017 Dec 15;9(19):15132-15143.
12. In line 576, ref. 68, reference should be correctly completed: J Parkinsons Dis. 2017;7(3):433-450.
13. In line 578, ref. 69, reference should be correctly completed: Autophagy 2018;14(4):717-718.
14. In line 587, ref.739, reference should be correctly completed Clin Chim Acta. 2016 May 1;456:107-114.
Author Response
The article summarizes the common trends among the genetic-associated cases of PD and AD. The authors indicate that these similarities could be of interest to find reliable biomarkers and efficacious therapies. Throughout the manuscript, the authors give some data, obtained from different research articles, in order to answer two questions: i) why some neurons are more vulnerable than others? and ii) are there common mechanism of neuronal death in patients inheriting mutations that lead to early-onset PD or early-onset AD?
The topic of the article covers the aim of the Special Issue "Genetics of Neurodegenerative Diseases: Focus on Progression and Response to Treatment”, presenting a cohesive picture of the state-of-the-art in the field, and helping to advance in understanding and management of neurodegenerative diseases. The manuscript describes a novel viewpoint and perspective, looking for similarities and differences in genetic between early-onset PD or early-onset AD.
Answer: thanks for the positive comment.
Although it is a valuable article, authors should provide the below changes:
1. Reference numbers should be placed in square brackets [], as it is indicated the Instructions for Authors of the Journal.
Answer: we appreciate the comment and, according to the information in the webpage, references have been formatted as it is indicated in the guidelines of the journal.
2. Size of the words in the draws of Figure 1 and 2 should be increased to improve the quality of the figures.
Answer: this good suggestion has been taken into account and, accordingly, the lettering in the draws of figures 1 and 2 were modified in order to improve quality and facilitate understanding.
3. In line 209, abbreviation should be included: “The K670M/N671L Swedish mutations (APPSwe) of the amyloid precursor…”, to define it for the first time because it is used afterwards.
Answer: we appreciate the suggestion, which is taken into account in the revised version of the manuscript.
4. In line 220, it should be clarified that in Tg2576 has been observed an overexpression of VDAC1 in hippocampus. Because in the present form, it seems that the transgenic mouse has been generated by the overexpression of VADC1 gene.
Answer: this good suggestion has been taken into account and this sentence has been rewritten accordingly.
5. In line 244, reference should be corrected, the correct is [50].
Answer: we appreciate the comment, which is taken into account in the revised version of the manuscript. This reference has been corrected. Thanks for noticing the mistake.
6. In line 288, the assertion is controversial. Authors should be clarified if this is an own opinion or it is a scientific convention, in which case authors must provide bibliographical references.
Answer: we appreciate the comment and after careful consideration we think that the sentence may be omitted (it does not appear in the revised version of the manuscript).
7. All the references should be noted as it is indicated in the guide for authors: 1. Author 1, A.B.; Author 2, C.D. Title of the article. Abbreviated Journal Name Year, Volume, page range. Available online: URL (accessed on Day Month Year).
Answer: we appreciate the comment and, accordingly, the list of references have been edited and hopefully it now fits with the requirements of the journal.
8. In line 469, ref. 32, it should be indicated [Epub ahead of print].
Answer: we appreciate the comment, which is taken into account in the revised version of the manuscript. Thanks for noticing the mistake.
9. In line 514, ref. 46, there is a typo, repetition of J.F.J.F.
Answer: we appreciate the comment, which is taken into account in the revised version of the manuscript. Thanks for noticing the mistake.
10. In line520, ref. 47, there is a typo: “A., M., C.-T., M., V., F., C., J., D. R., and D., F.”
Answer: we appreciate the comment, which is taken into account in the revised version of the manuscript. Thanks for noticing the mistake.
11. In line 539, ref. 54, reference should be corrected: Oncotarget. 2017 Dec 15;9(19):15132-15143.
Answer: we appreciate the comment, which is taken into account in the revised version of the manuscript. Thanks for noticing the mistake.
12. In line 576, ref. 68, reference should be correctly completed: J Parkinsons Dis. 2017;7(3):433-450.
Answer: we appreciate the comment, which is taken into account in the revised version of the manuscript. Thanks for noticing the mistake.
13. In line 578, ref. 69, reference should be correctly completed: Autophagy 2018;14(4):717-718.
Answer: we appreciate the comment, which is taken into account in the revised version of the manuscript. Thanks for noticing the mistake.
14. In line 587, ref.739, reference should be correctly completed Clin Chim Acta. 2016 May 1;456:107-114.
Answer: we appreciate the comment, which is taken into account in the revised version of the manuscript. Thanks for noticing the mistake.
Observation: major changes are highlighted in yellow. Grammatical corrections and deletions are not highlighted.
Reviewer 3 Report
In the paper titled “Lessons on differential neuronal-death-vulnerability from familial cases of Parkinson’s and Alzheimer’s diseases” the authors tried to summarize the common mechanistic features between genetic-associated cases of Parkinson’s disease and Alzheimer’s disease. The topic chosen for the review is very interesting and the abstract is concise and clear.
But, the authors have failed to comprehensively summarize the known facts/details on the mutated genes in the familial cases of PD or AD. Also, they failed in drawing conclusions on the common cellular mechanisms/features altered between the PD and AD. For most part, authors give references to previous reviews or papers instead of briefly summarizing on them. Authors have drawn very broad conclusions and the connection between the known facts and the arguments put forward therein do not clearly convey the message.
Overall, the review is very poorly written and failed to deliver the information one would anticipate to obtain, especially, after reading at the very well-written abstract. Otherwise, it would have been a good addition to the field.
Therefore, I regret to mention that I do not recommend the review for consideration towards publication in IJMS.
Author Response
In the paper titled “Lessons on differential neuronal-death-vulnerability from familial cases of Parkinson’s and Alzheimer’s diseases” the authors tried to summarize the common mechanistic features between genetic-associated cases of Parkinson’s disease and Alzheimer’s disease. The topic chosen for the review is very interesting and the abstract is concise and clear.
Answer: thanks for the positive comment.
But, the authors have failed to comprehensively summarize the known facts/details on the mutated genes in the familial cases of PD or AD.
Answer: we are afraid but the aim of the paper has never been the one commented by the reviewer.
Also, they failed in drawing conclusions on the common cellular mechanisms/features altered between the PD and AD.
Answer: we appreciate the comment that was also raised by another reviewer. On the one hand, and in our opinion, this issue has been fixed in the revised version. On the other hand, if the reviewer refers to protein aggregation that i) does not always course with AD/dementia and ii) therapy addressed to “solve” this aggregation has, much unfortunately, failed; the article focuses on alternative paths must be explored. The shift in PD therapy towards protein aggregation, that has failed in AD, does not seem very logical to us. In addition, when we started this project we already knew that it is virtually impossible to explore all the possible/potential links between AD and PD in a paper.
For most part, authors give references to previous reviews or papers instead of briefly summarizing on them. Authors have drawn very broad conclusions and the connection between the known facts and the arguments put forward therein do not clearly convey the message.
Answer: this is a commissioned paper and we thought very hard before start writing it. In our opinion, the paper provides new relevant information in form of AD-PD connections that are usually overlooked. We agreed upon submission (probably written in the cover letter) that the paper could be improved with the input from reviewers. In this sense, we have no problem in recognizing that the revised version is much better than the first.
Overall, the review is very poorly written and failed to deliver the information one would anticipate to obtain, especially, after reading at the very well-written abstract. Otherwise, it would have been a good addition to the field.
Answer: we are sad to read that the information the reviewer anticipated was not in the manuscript. In our opinion and, we understand that also in the opinion of the other two reviewers.
Observation: major changes are highlighted in yellow. Grammatical corrections and deletions are not highlighted.
Round 2
Reviewer 3 Report
I recommend this article for publication.
There are few grammatical errors that needs to be checked properly and formatted.